# Navigating Legal and Regulatory Frameworks to Achieve the Resilience and Sustainability of Indigenous Socioecological Systems

**Stephen Chitengi Sakapaji** [1,*], **Jorge García Molinos** [1], **Varvara Parilova** [2], **Tuyara Gavrilyeva** [3] and **Natalia Yakovleva** [4]

1 Arctic Research Center, Hokkaido University, Sapporo 001-0021, Japan; jorgegmolinos@arc.hokudai.ac.jp
2 Graduate School of Environmental Studies, Tohoku University, Sendai 980-0845, Japan; varvaraparilova@gmail.com
3 Institute of Engineering and Technology, North-Eastern Federal University, 677027 Yakutsk, Russia; tuyara@list.ru
4 KEDGE Business School, 75012 Paris, France; natalia.yakovleva@kedgebs.com
* Correspondence: stevesakapaji@yahoo.com; Tel.: +81-70-1556-1986

**Abstract:** The sustainability of Indigenous Socioecological Systems (ISES) largely depends on well-crafted policy regulations. In particular, Indigenous traditional food systems (ITFS) are an essential component of ISES that provide a variety of culturally accepted, healthy foods while also playing an important role in cultural, spiritual, and economic value to the Indigenous people (IP). Thus, sustainably managing these traditional natural resources must be a priority. As custodians of much of the world's ecological system, IP have, for generations, exhibited sustainable lifestyles in governing these systems. However, Indigenous perspectives and voices have not been properly reflected in the ISES sustainability discourse, and few comparative case studies have addressed this issue. This study contributes to fill this research gap using a desktop research method based on the Political Ecological Theoretical Framework (PETF) to examine how existing regulatory policies may affect the resilience and sustainability of ISES-ITFS, especially in relation to growing environmental and climatic pressures. Two Indigenous communities, the Karen in Thailand and different Indigenous groups in the Republic of Sakha (Yakutia) in Russia, are examined as case studies. Our study provides crucial insight that should help the development of robust policy interventions that integrate Indigenous concerns into policies and regulations, emphasizing self-determination, cultural preservation, and land rights. The findings emphasize the necessity for comprehensive legal frameworks prioritizing Indigenous involvement and concerns in climate and sustainability policy implementations. The ultimate goal is to foster meaningful dialogues between policymakers and IP in navigating the climate and sustainability challenges of our time.

**Keywords:** Indigenous; people; Karen; Yakutia; resilience; sustainability; policy; legal framework

## 1. Introduction

Indigenous people (IP) worldwide have been known to maintain a close bond with their land, territories, and resources. In this research paper, we collectively refer to this intimate bond as the Indigenous socioecological system (ISES), and hereby define it as a linked system of "people and nature" [1] that encompasses the interactions between Indigenous peoples and their local surrounding environment [2,3]. It includes natural resources such as water, land, and wildlife, which these people depend on. This also encompasses the spiritual and cultural aspects attached to these resources. Because of IP's wisdom and advanced resource management methodologies amassed over generations, the ISES are frequently distinguished by their enduring sustainability [4,5]. Within this discourse, here we focus on the Indigenous traditional food systems (ITFS) as a fundamental

key component of ISES. ITFS can be broadly defined as the culturally acceptable foods that are produced or procured from the local natural environment [6]. Natural resources, which are the fundamental building blocks for Indigenous traditional culinary practices and diets, are of paramount importance for the procurement of indigenous food [7,8]. Indigenous communities have relied on the local environment for generations to meet their nutritional requirements, as it provides a rich variety of aquatic and terrestrial species, fauna, and flora. Additionally, indigenous food systems frequently stand out for having a thorough approach to food procurement that respects and uses each element of the ecosystem. These practices encompass not just food consumption, but also other uses provided by food species, such as clothing or traditional medicine, as well as their cultural connotations that fortify the ecological and spiritual importance of natural resources. Because ITFS include all interlinked actors and activities involved in the processing, distribution, consumption, and disposal of foods, and because they are influenced by a complex matrix of interacting socioeconomic and environmental factors, the implementation of inclusive, participatory, and fair regulatory frameworks is essential to their long-term sustainability [9].

A vital concept when delving into a discussion of the relationship between IP and their socioecological systems is that of kincentric ecology [10–13]. The concept hinges on the view that human beings are essential components of a broader ecological network, thereby redefining their function as responsible custodians and engaged participants in the intricate web of life [14,15]. Key in the concept of kincentric is an intense reverence for entities other than humans. In stark contrast with the prevailing anthropocentric ideologies of Western societies, where nature is viewed as a simple resource that should be exploited, kincentric ecology fosters an attitude of respect and admiration for the natural environment. This view also regards all things, including water bodies, forests, air, the sun, plants, and animals, as possessing some level of consciousness capable of directing human life [15,16]. For IPs, all organisms are regarded as relatives possessing distinct agency and consciousness. From this viewpoint, one can conclude that the sustainable lifestyles exhibited in these societies are a result of this deep connection to their natural environment [16]. Thus, the kincentric ecology framework is grounded upon the fundamental values of stewardship and responsibility, which establish the relationship between Indigenous communities and their natural surroundings. Sustainable resource management is a key component that naturally arises when discussing the concept of kincentric, which entails safeguarding the land's well-being and continued sustenance for future generations [16].

In discussing the sustainability and resilience of the ISES, and ITFS in particular, policy and legal frameworks have a profound impact on the trajectory of these systems. Legal and regulatory frameworks can either facilitate or hinder the ability of Indigenous communities to address the consequences of global environmental and climate changes and maintain their sustainable lifestyles [16–19], supporting the long-term viability of ISES and ITFS as well as the continuation of customary methods, which frequently stand out for their low carbon emissions and increased ecological efficiency. Throughout history, Indigenous societies have faced challenges related to marginalization and exclusion from decision-making processes, which have resulted in the development of policies that fail to consider their unique needs and priorities [19–22]. It is thus imperative that legal frameworks recognize and respect the unique knowledge, rights, and customs of Indigenous communities across the globe. However, evidence demonstrates that the formulation of policies, especially those related to the sustainability and resilience of ISES and ITFS, is often done without the engagement and participation of Indigenous communities despite their vast and valuable knowledge in sustainably managing their lands, resources, and territories [18–24].

Given this context, the impact of policy and legal structures on the sustainability and resilience of the ISES-ITFS is, without doubt, a critical factor to be considered when attempting to resolve the challenges that Indigenous communities are confronted with because of environmental and climate changes. Similarly, few comparative research studies have directly addressed the potential impact of regulatory and legal frameworks on the sustainability and resilience of ISES and ITFSs [25–27]. While the diversity of IPs compli-

cates the comparison and generalization of the effects of legal frameworks on ISES-ITFS, the political sensitivity of these matters often discourages or restricts researchers from addressing them. Moreover, the absence of global cooperation among scholars from various nations poses a barrier to conducting comparative research on the topic [28–32]. This situation significantly restricts our ability to identify effective mechanisms that support Indigenous rights, inclusivity, and participation in the creation and implementation of legal and regulatory frameworks. Accordingly, in this study, we contribute to filling this research gap using an innovative comparative case-study methodology to thoroughly examine the unique opportunities and obstacles that legal and regulatory frameworks can pose for Indigenous communities as they endeavor to mitigate and adapt to the consequences of climate and environmental change within their ISES-ITFS.

A clear understanding of the relevance of ITFS to the IP and their role within the ISES is needed as a prerequisite to fully examine the intricacies surrounding the legal and regulatory frameworks and the impact these may have on the sustainability of ISES and ITFS [33]. ITFS comprise complex interconnections among individuals, their surroundings, and sources of sustenance, which transcend the concept of basic nutrition [10–13]. ITFS, which hinges on accumulated traditional ecological knowledge (TEK), incorporates spiritual beliefs associated with food sources, sustainable harvesting techniques, and ecological awareness [29]. Through communal dining and food-related ceremonies, these systems promote cultural identity and social cohesion, thereby strengthening community ties and transmitting traditional values [15–18]. ITFS is known to place a high value on fostering a mutually beneficial association with the natural world by conducting sustainable resource management activities such as controlled hunting and rotational farming, which effectively safeguard against the overexploitation of local biodiversity.

ITFS are also known for their capacity to adapt to natural variations in local conditions and enhance environmental change resilience [16–19]. Furthermore, in stark contrast with industrialized food systems, ITFS provides nutrition-dense, well-balanced diets by utilizing a variety of food sources, such as wild meat and vegetation, cultivated crops, and the use of traditional processing methods like fermentation [30]. However, historical injustices, rapid environmental and climate changes, and the allure of modernization pose growing threats to the sustainability of ITFS. Therefore, it is critical to acknowledge the significance of ITFS, given that they serve as paradigms for sustainable food production, foster social and environmental resilience, advance food security and health, and safeguard cultural heritage and knowledge for posterity [15–19]. In summary, in this paper, we focus on ISES and ITFS for the following four reasons:

a. These systems frequently incorporate locally adapted and sustainable practices that have been developed for generations, thereby enhancing the resilience and sustainability of dependent IPs [17,18].

b. Advocating for policy instruments that acknowledge and bolster ITFS has the potential to improve food security among vulnerable communities such as IP and promote varied, healthy, and culture-based diets [9].

c. ISES-ITFS are instrumental in upholding community identity and preserving cultural heritage [18]. Thus, policy measures that safeguard and advance these systems have the potential to promote economic and social welfare among Indigenous communities, thereby making a positive contribution to the overarching objectives of social justice and equity [18].

d. Lastly, Indigenous food practices are known to be in harmony with nature and align with global initiatives such as the 2015 Paris Climate Accord and the United Nations (UN) Agenda 2030 for Sustainable Development [18].

Therefore, to successfully promote climate adaptation, resilience-building, and sustainability for the ITFS while traversing the regulatory and legal landscapes, it is essential to have a thorough understanding of the unique challenges and opportunities Indigenous communities face. Conversely, an examination and evaluation of current regulatory and legal structures pertaining to the promotion of climate adaptation, resilience-building, and

the sustainability of ITFS can yield valuable knowledge regarding the development of effective policies that empower Indigenous communities to withstand the effects of climate change [20,28]. Moreover, this can provide valuable insights for international endeavors and pledges to address climate change through the promotion of sustainable behaviors and the achievement of sustainable development objectives.

Considering all these factors, this research paper endeavors to analyze the impact of legal and regulatory policies on ISESs to advance ways through which such policies can effectively strengthen the traditional practices, wisdom, and rights of IP while also exploring the adverse consequences of inadequately crafted legal frameworks that have their origins in historical marginalization prejudice and superiority. We further examine the potential for inclusive and culturally sensitive policies that integrate Indigenous perspectives, fostering collaboration and partnerships among diverse stakeholders to develop policies that ensure the continuity of ITFS and the sustainability of ISES. Ultimately, this study endeavors to underline the broader implications of well-crafted policy and legal frameworks in sustaining ISES-ITFS, while preserving ecologically efficient traditional practices with a low carbon footprint. The following are the main objectives that serve as the guiding pillars of this comprehensive research paper:

- Investigate the impact of existing regulatory and legal frameworks on the adaptive capacity, resilience-building, and sustainability of Indigenous Socioecological Systems (ISES), with a specific focus on Indigenous traditional food Systems (ITFS).
- Establish a theoretical model for the sustainability and resilience of the ITFS to facilitate the integration of IP's concerns and voices into contemporary policies, and legal, and regulatory frameworks.
- Foster dialogue among Indigenous communities, policymakers, and stakeholders, to safeguard and reinforce the rights and sustainability of ISES, particularly in the face of accelerating climate change and widespread environmental exploitation.

## 2. Methodology

We employ a comparative case study approach to investigate the intricate interplay between legal and regulatory frameworks within Indigenous communities from two distinct geographical and socio-cultural settings: Indigenous communities of Yakutia (Sakha Republic, Russia), including multiple ethnic groups, and the Karen Indigenous People in Thailand (Figure 1). This approach has been chosen to enable a comprehensive exploration of the multifaceted dynamics that arise from the intersection of legal and regulatory constructs across different socio-cultural and geographical contexts. To this end, we make combined use of desktop research methodology analysis with a focus on existing literature reviews on the topic (38% of the total documents revised listed in the reference list of this paper), national legal and regulatory laws governing the Indigenous territories in our study communities (48%), and comprehensive analysis of international Indigenous rights reports (14%). This methodological framework is strategically designed to discern, in a highly refined manner, the nuances of challenges, opportunities, and ultimate outcomes arising from the presence of legal and regulatory frameworks within these unique Indigenous communities. By juxtaposing and analyzing these distinct cases, this comparative inquiry aspires to furnish a comprehensive and insightful understanding of the complex ways in which these legal and regulatory frameworks may shape and influence climate adaptation, resilience enhancement, and sustainable development initiatives within distinctive ISES-ITFS, spanning diverse cultural and geographical contexts.

Using the sources of information described above, this research paper develops a Political Ecological Theoretical Framework (PETF) model to analyze and examine the role of regulatory and legal frameworks in the adaptation and sustainability of the ISES-ITFS (Figure 2). Although the concepts of the PETF have been used before in other fields of environmental research, such as conservation ecology [34] and waste management [35], its application to Indigenous socioecological systems and traditional food systems has, to the best of our knowledge, not been attempted before. Political ecology examines the

intricate interactions between political, economic, social, and environmental factors that shape resource use, distribution, and access [34–38]. Applied to our study, Political ecology provides a lens through which to analyze the power dynamics, sociopolitical contexts, and environmental implications that can influence the adaptation, resilience-building, and sustainability efforts within Indigenous socioecological systems. The implementation of this methodological framework allows us to unravel the intricacies emerging from the existence of legal and regulatory frameworks and structures within our study communities.

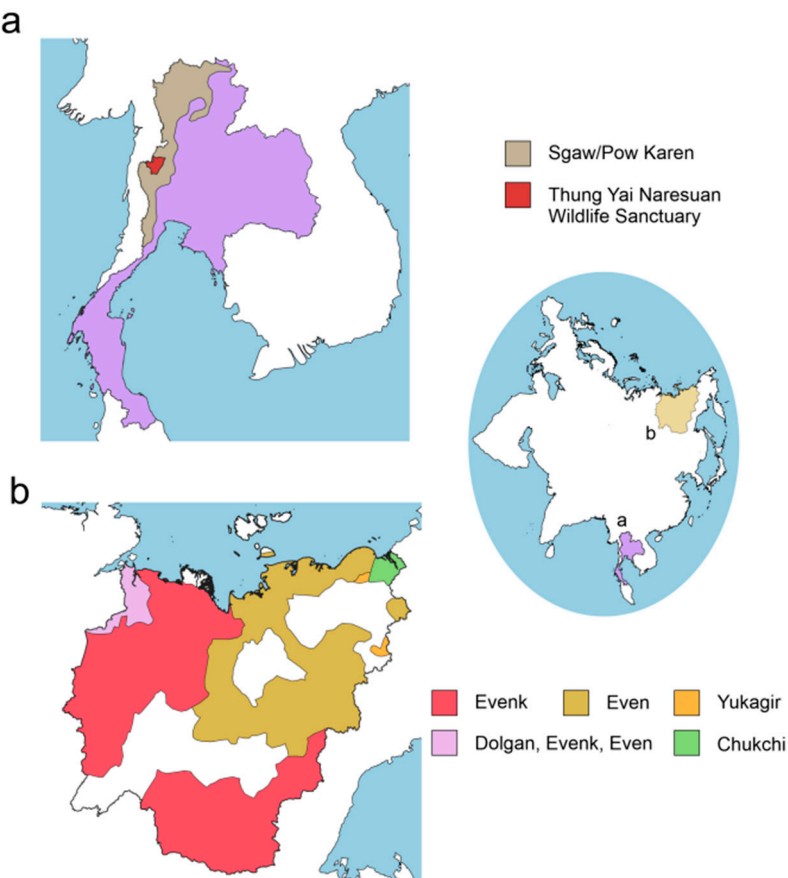

**Figure 1.** Location of our case studies in (**a**) the Sakha Republic and (**b**) Thailand. The maps provide approximate distributions of (**a**) the Karen People in Thailand and (**b**) the main Indigenous minority Peoples of the North, Siberia, and the Far East in the Sakha Republic. Panel (**a**) also depicts the location of the Thung Yai Naresuan Wildlife Sanctuary where the Sanephong and Koh Sadueng Karen communities discussed in the text are located. White areas of the Sakha Republic in (**b**) are ethnically dominated by the Yakuts (Sakha People), a large Turkish ethnic group [33].

The development of the PETF model in the context of this study is particularly relevant for assessing the influence of regulatory and legal frameworks on ISES and ITF. The PETF has also the capacity to unravel the underlying power dynamics within the sociopolitical and environmental contexts, unraveling the implications for the Indigenous communities to adapt, build resilience, and ensure the sustainability of their ITFS and ISES as a whole. Thus, the PETF model approach paves the way for a thorough assessment of historical marginalization, unequal resource distribution, and Indigenous rights within the framework of climate resilience and sustainability [35–37]. Therefore, the ultimate goal of this model is to inform effective mechanisms for more equitable and regulatory policy instruments that align with the values and needs of Indigenous communities while promoting climate resilience and sustainability within ISESs.

Below, we discuss three areas where the PETF can be effectively utilized in the context of ISES-ITFS as applied in our study:

- Indigenous rights and power dynamics: Political ecology plays an important role in facilitating the assessment of power dynamics that exist within legal and regulatory frameworks [35–39]. Concerning Indigenous communities' efforts to assert their rights over their territories, resources, and traditional knowledge, power dynamics may either empower or disempower these communities. Thus, political ecology has the potential to facilitate a scholarly analysis of how legal structures can either support or contest the historical inequities and wrongdoings that Indigenous peoples have endured [38,39].

- Sociopolitical contexts and environmental implications: This PETF model has the potential to equip users with the means to evaluate the sociopolitical contexts that influence the formulation of legal and regulatory decisions [38,39]. The prospective effects of these decisions on the ecological integrity of ISES are considered as they pertain to land use, environmental policies, and the management of natural resources. A proper understanding of this matter is fundamental in assessing the efficacy and equity of legal structures in their pursuit of climate resilience and sustainability.

- Unequal Resource Distribution and Historical Marginalization: Lastly, political ecology permits smooth resource distribution, especially in the context of ISES-ITFS. Furthermore, political ecology helps to fully examine how regulatory and legal frameworks have contributed to the historical marginalization of Indigenous communities and the perpetuation of resource inequities including land rights [36–39]. This is a crucial standpoint for advocating policies that rectify these historical injustices.

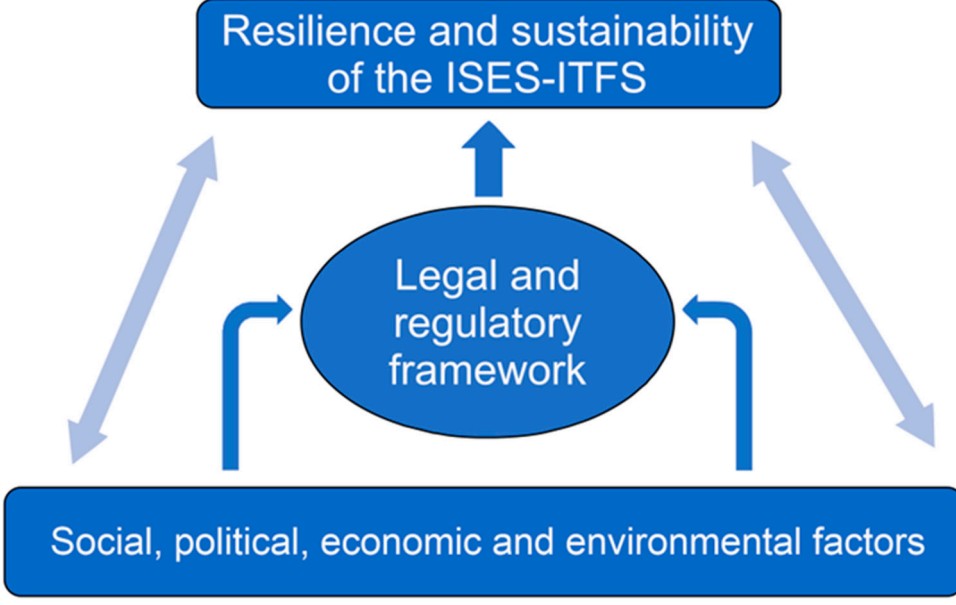

**Figure 2.** Conceptual diagram showing the Political Ecological Theoretical Framework model in the context of our research paper. Social, political, economic, and environmental factors defining the dimensions of the interaction between IP and other actors (national and regional regulatory bodies, industries, research institutions…) shape the formation and implementation of legal and regulatory frameworks that can impact (positively or negatively) the resilience and sustainability of the ISES and ITFS. In the context of this study, over and above the direct effects of these dimensions on ISES-ITFS (grey arrows), we focus on the legal and regulatory framework as an instrument of power channelizing and articulating the effects of the different dimensions on the ISES-ITFS (blue arrows). The double head of the grey arrows symbolizes the possibility for IP to exert power on the legal and regulatory system through their actions and agency (e.g., litigation, political representation, public awareness) on all or some of these dimensions.

### 2.1. Power and Power Dynamics as Used in Our Study

Within the domain of legal and regulatory frameworks, power broadly pertains to the capability or aptitude of institutions, groups, or individuals to exert influence, mold, or command over the formulation, implementation, and interpretation of policies, laws, and regulations [39]. In the present context, power dynamics encompass the allocation, utilization, and bargaining of authority among governmental entities and regulatory bodies in our case studies. In this study, we view power dynamics as comprising formal or informal, hierarchical or horizontal relationships of authority, control, and influence in natural resource management and conservation. We understand that comprehending power dynamics is critical for scrutinizing the processes by which legal and regulatory decisions are formulated, identifying the beneficiaries and detriments of such decisions, and examining how power structures can either sustain or contest societal inequities, injustices, and systemic problems [39]. In our study, we are specifically focusing on how power and power dynamics (legal and regulatory instruments in Figure 2) can influence the resilience and sustainability of ISES and ITFS.

### 2.2. Recognizing the Role of Innovative Legal and Regulatory Frameworks in Shaping the Sustainability and Resilience of ISES and ITFS

Regulatory and legal policy frameworks that are appropriately crafted have the potential and capacity to significantly influence the long-term viability of ISES and ITFS [38,39]. The intricate and interconnected networks of social, cultural, economic, and ecological elements that comprise the ISES-ITFS are profoundly influenced by the relationship between Indigenous communities and their environments, as already discussed in the introduction section. To guarantee the long-term viability of the ISESs and ITFSs, it is vital to fully understand and comprehend the ramifications of deliberate and meticulously crafted legal and regulatory frameworks.

Below, we outline and discuss areas that are necessary for the development and implementation of novel regulatory structures that adhere to globally acknowledged norms and incorporate the perspectives, rights, and liberties of Indigenous communities regarding access to their ancestral territories.

i.  Land tenure and resource rights: The ability of Indigenous communities to effectively govern and protect the sustainability and resilience of their ISES and ITFS is heavily reliant on the secure tenure of land and resource rights [38,39]. Indigenous land rights must be legally recognized to establish a sustainable framework for resource management. In the past, land tenure and resource rights of the IP have frequently been compromised or disregarded, resulting in the exploitation of resources and the degradation of the ISES and ITFS. Improper and unstructured legal instruments that do not acknowledge and safeguard the land rights of IP have the potential to undermine the IP capacity to manage their resources and territories in a sustainable manner [40–43]. The United Nations Declaration on the Rights of Indigenous Peoples (UNDRIP) emphasizes the rights to lands, territories, and resources of the IP [41]. In addition, its emphasis on Free, Prior, and Informed Consent (FPIC) concerning decision-making processes has an impact on Indigenous communities. Land tenure and resource rights as enshrined in the UNDRIP provide a solid basis for the sustainable management of ITFS and ISES as a whole [41].

ii. Traditional ecological knowledge (TEK) protection: TEK is a "cumulative body of knowledge, practice, and belief, evolving by adaptive processes and handed down through generations by cultural transmission, about the relationship living beings (including humans) with one another and with their environment" [19]. Because TEK is a way of knowing that builds on local experience and adapts to changes [17–19], it Is highly relevant to resilience and adaptation against current climate change impacts and continuous environmental changes especially when it comes to the sustainability of the ISES and ITFS [18,19]. Given this importance, legal frameworks that focus on the protection of TEK, including intellectual property rights for Indigenous knowledge

holders, are crucial for promoting the continued use of traditional practices that contribute to ISES and ITFS sustainability. This reality is recognized by the Convention on Biological Diversity (CBD), which acknowledges the importance of Indigenous and local communities' traditional knowledge, innovations, and practices for the conservation and sustainable use of biological diversity [43]. In addition, the rights to genetic resources and the just and equitable distribution of benefits resulting from their use are explicitly addressed in the Nagoya Protocol, which serves as an additional treaty to the CBD and safeguards ITFS.

iii. Traditional hunting, gathering, and fishing activities: In many Indigenous communities hunting and fishing practices are notably impacted by legal and regulatory structures that seem to protect the environment but, in doing so, harm the very existence of IP and their SES and TFS. Recently, the conservation of wildlife resources and biodiversity has garnered considerable international attention. Government policies designed to preserve these natural resources are posing challenges to the socio-economic activities of IP, notably their gathering, fishing, and hunting practices. For example, in the Sakha Republic, the regulation of hunting follows an interdepartmental structure where the Ministry of Ecology, Nature Management, and Forestry of the Republic has the regulatory and administrative power, while the Department of Hunting and Specially Protected Territories is entrusted with the organization of activities [44]. Despite some positive recent advancements, this complex regulatory structure poses important coordination challenges that ultimately impact the equitable and fair access of the Sakha IPs to hunting in the Republic [44]. Issues include the insufficient implementation of priority hunting rights for small Indigenous peoples and inadequate and dysfunctional legislation for compensation of damages inflicted on hunting resources by extractive industries impacting the habitats of wild animals. This example illustrates how inadequate policies can severely curtail the traditional way of life of IPs in a region. Therefore, when properly designed, legal and regulatory frameworks can foster the sustainability of ISES-ITFS by recognizing and respecting Indigenous rights, integrating TEK into conservation strategies, and promoting adaptive management practices [45]. To this end, a regulatory focus on sustainable practices that incorporate cultural, socioeconomic, and environmental considerations through the establishment of appropriate measures such as territorial use rights, quotas, or seasonal restrictions aligned with traditional practices, can promote ecosystem resilience and the long-term viability of Indigenous hunting and fishing practices [45].

iv. Conservation and environmental management. Legal and regulatory frameworks have the potential to impact conservation initiatives and environmental management within the ISES-ITFS if not properly crafted. Similarly, conservation efforts conducted by IPs may be facilitated or impeded by these frameworks. Therefore, consensus-building processes regarding conservation and resource management are more likely to produce enduring results when Indigenous communities are engaged in collaborative efforts [46,47]. An increase in environmental stewardship may result from the legal recognition of co-management arrangements in which Indigenous communities participate as equal participants [46,47]. Moreover, collaborations and partnerships among governments, non-governmental organizations (NGOs), and Indigenous communities can be improved by legal and regulatory frameworks [46]. Indigenous-led sustainability initiatives may be supported by such alliances, which may also provide vital funding, technical advice, and resources. For example, in New Zealand, the Māori Resource Management Act of 1991 grants Māoris a substantial influence in the governance of natural resources situated on their ancestral lands [48]. Effective protection of Māori cultural heritage and environmental values accomplished over the years demonstrates the success of the act. For example, the Wairau River Agreement, signed in 1999 between the New Zealand government and seven Māori iwi or tribes, has enhanced the river's quality and protected its biodiversity under joint collaborative management [48]. Many other examples of enhanced resilience and

sustainability of both the ISES and the ITFS achieved through the implementation of innovative legal and regulatory frameworks are available worldwide. In Australia, the Indigenous Land Rights Act of 1993 bestows the capacity to assert native title claims over ancestral territories upon Aboriginal and Torres Strait Islander communities [49]. Murray Island in the Torres Strait was granted native title rights to the Meriam people with the landmark Mabo decision rendered by the High Court of Australia in 1992. Similarly, the 1997 Indigenous Peoples' Rights Act (Philippines) is another important act acknowledging the territorial sovereignty and self-governance rights of Indigenous peoples such as, for example, the recognition of the Dumagat people's ancestral domain in the Sierra Madre mountains [50]. These successful examples demonstrate how Indigenous participation in the sustainable management of their traditional territories and resources can be ensured by effective legal and regulatory frameworks that safeguard the cultural heritage and traditions of Indigenous communities and contribute to the environmental preservation of these regions. By adopting such initiatives, governments and Indigenous peoples can foster a more equitable and fair relationship of mutual benefit, while also safeguarding the environment and Indigenous rights and cultures. Nonetheless, despite the unquestionable progress made on many fronts, IPs today are still subject to widespread inequalities, power struggles, and rights violations across the world. Similarly, in the context of ISES-ITFS, there is still much to be done to achieve the required status quo of environmental justice and legal recognition for IPs that will secure their rightful claims to manage and decide on the use of their ancestral lands and natural resources within.

## 3. Case Study 1: The Karen Indigenous People

### 3.1. Historical Background

The Karen Indigenous people of Thailand are among the nine Indigenous ethnic groups that have received official recognition from the country. It is postulated that their migration from Tibet or Mongolia to Myanmar and Thailand occurred via China [51]. The Karen Indigenous people's history in Thailand is an enthralling narrative that spans more than a millennium. It is distinguished by their protracted migration from distant territories to the area and the development of their unique way of life [51]. The Karen people, who are thought to have settled in the region around a millennium ago, have significantly influenced the cultural fabric of northern and northwestern Thailand. The Karen, a people originating from regions far beyond the borders of Thailand, undertook a profound and life-altering expedition motivated by an assortment of factors. These encompassed the desire to acquire arable land, seek refuge from political instability in their countries of origin, and explore new prospects in an unfamiliar territory [52,53]. They established communities in the dense forests and highlands of northern and northwestern Thailand, which later became their ancestral abodes [53]. In addition to providing them with the essential resources required for survival, these lands also functioned as the backdrop against which their unique cultural identity was etched.

The Karen Indigenous population in Thailand comprises four distinct subgroups: Sgaw, Pwo, Kayah, and Toungthu Karen. The Sgaw and Pwo Karen constitute approximately 70% and 25% of the total Thai Karen population, respectively. Here, we concentrate on two Pwo communities, Sanephong and Koh Sadueng, situated in the Laiwo subdistrict of Kanchanaburi province (Figure 1a). These communities are surrounded by the Tanowsri mountain range; a hilly terrain comprising numerous narrow valleys that serves as a natural demarcation line between Thailand and Myanmar. It is hypothesized that Pwo Karen migrated to this region from China in the 13th century [54,55]. These tenacious individuals are bestowed with a cultural heritage that is intricately linked to the environment. Their rituals, beliefs, and sustainable circular shifting agricultural practices are deeply intertwined with nature and continue to endure. Traditional agricultural systems and means of subsistence continue to provide Pow families and communities with vital resources [54,55].

The Karen people's traditional way of life in these areas has evolved over generations into a complex interweaving with the natural surroundings refining their skills in foraging, gathering, and agriculture by capitalizing on their extensive knowledge of the surrounding ecosystems [56–59]. As a result, their traditional food systems, firmly grounded in the natural cycles, are extremely diverse and demonstrate their intimate knowledge and relationship with the land [60]. For example, reflecting the extreme diversity of local food available to the community, the Sanephong TFS has been described to include 387 known local traditional food species of plants and animals between wild harvesting and the cultivation of traditional crops [57]. Over the course of several centuries, the Karen people have diligently maintained their cultural legacy by transmitting their languages, customs, and belief systems across successive generations. The lasting impact of this heritage has significantly enriched the contemporary cultural fabric of Thailand, serving as a testament to the Indigenous peoples' ability to endure and thrive in a variety of environments. The illustrious history of the Karen people in Thailand provides a compelling illustration of the eternal bond that exists between humans and their surroundings. This statement emphasizes the criticality of acknowledging and preserving the cultural legacy of Indigenous populations, while simultaneously confronting the modern obstacles they face in a dynamic global landscape.

### 3.2. The Introduction of Legal and Regulatory Framework—Colonial Era

The onset of the colonial era in the 19th century marked a turning point in the history of the Karen people and their ancestral territories. During this time, the Thai kingdom expanded its authority and control over the Karen territories bringing a novel era marked by significant social, cultural, and political transformations [58]. The territorial expansion and power consolidation by the Thai government crystallized in a succession of assimilation policies aimed at suppressing the Karen language, customs, and religious practices [58,59].

Similarly, the advent of land policies in the colonial targeted the traditional Karen territories, which had historically supported their communities for millennia, in an effort to seize control of their valuable resources [59,60]. Land confiscations and reassignments led to widespread displacement, loss of livelihoods, and social upheaval among the Karen communities [59–61]. Movements of resistance by the Karen people against these discriminatory policies have emerged to safeguard their territory, dialect, and customary methods, frequently confronting formidable odds. These conflicts over land rights and cultural preservation would come to define the history of the Karen people.

### 3.3. Legal and Regulatory Framework—Post-Colonial Era

Thailand's independence from French colonial rule in 1949 brought a period of continued challenges and discrimination for the Karen people [62,63], during which they continued to be subjected to marginalization and coerced displacement by the Thai government [60]. This pattern served to escalate existing animosities between the Karen communities and the central government. This situation did not undergo any substantial transformation following the country's recent independence [63]. Karen communities continued to be subjected to discriminatory practices and systemic disadvantages in access to their land resources, education, health care, and economic opportunities, broadening socioeconomic disparities between the Karen and the Thai society as a whole [64]. A significant concern that arose in the aftermath of colonialism was the forced displacement of a considerable number of the Karen population from their customary regions to low-lying areas [62–65]. The ramifications of this policy, implemented in the name of progress and modernization, were significant for Karen communities. Their centuries-old agrarian practices were disrupted by forced relocations, resulting in the loss of land, means of subsistence, and cultural ties to their ancestral territories [66]. Numerous Karen households experienced displacement and encountered difficulties in acclimating to unfamiliar lowland surroundings [67]. These forced relocations severely deteriorated the already strained relations between the Karen and the Thai government generating profound animosity and

hostility. Discontent grew as the Thai government's actions were perceived as a threat to their existence and a violation of their fundamental rights. Conflict and sporadic insurgent movements followed [65–67].

*3.4. Implications of Existing Legal and Regulatory Frameworks on the ISES and TFS of the Karen Indigenous Communities*

Throughout history, the Karen people have encountered limitations on their land rights because the Royal Thai government designated a significant portion of their ancestral territories as public land [63–67]. In this section, we examine instances of legal and regulatory frameworks that have affected the ITFS of the two study Karen communities (Sanephong and Koh Sadueng). By the late 1930s, the Royal Thai government was increasingly motivated to implement policies and legislation intended to preserve forests and nature in response to Karen's tenacity for autonomy in the use of natural resources within their lands. Presenting this free will as a fundamental risk to the conservation of natural resources, the Thai government enacted the Forest Reserve Act in 1941 to designate vast areas as national forests [65,66]. This meant that these Karen communities were no longer permitted access to vital hunting, gathering, and agricultural grounds, effectively disregarding traditional land tenure systems and prioritizing commercial forestry over traditional Karen practices.

Similarly, the 1961 Land Code Act confronted communal land-holding traditions curtailing the Karen's capacity to manage their territories collectively for traditional food production. The problem was worsened by concessions granted for mining and large-scale agriculture that caused large disturbances to the ecosystems supporting the Karen TFS. The persistent disregard for customary laws and governance structures combined with very limited participation in decision-making poses a substantial obstacle to the capacity of these Karen communities to manage their territory sustainably [66].

The recent implementation of legislation pertaining to water resource management and biodiversity conservation adds an extra layer of complexity by limiting the free access of the Karen communities to water resources and imposing limitations on traditional activities such as hunting and swidden agriculture. This situation is particularly tense for the Karen Indigenous communities within the Thung Yai Naresuan Wildlife Sanctuary (TYNWS), including Sanephong and Koh Sadueng. Since its inception in 1974, the TYNWS has seen numerous confrontations and disputes between the Karen communities and the Natural Park authority resulting from the complex interplay of socio-cultural dynamics and the implementation of conservation policies that restrict many of the Karen traditional practices such as their ancestral swidden agriculture, or the hunting, fishing, and harvesting of many wild animals and plants that form part of their ITFS [65,66]. These conservation laws often systematically disregard the Karen customary land tenure systems and traditional resource management practices [66]. Subsequent clashes over land use and resource access eventually led to forced evictions and the displacement of many of the formerly thriving Karen communities from within the sanctuary. This historical trajectory underscores the complex challenges at the intersection of conservation efforts and Indigenous rights within the Thung Yai Naresuan Wildlife Sanctuary [67].

In addition, the Karen people have been unjustly singled out for employing land clearance-intensive traditional agricultural techniques in violation of the 2017 legislation that criminalizes deforestation [63,66]. This has resulted in arrests and incarceration, further impacting their livelihoods given their direct reliance on the forests. In addition, the requirement of government-appointed village headmen by a law enacted in 2018 further undermines Karen's self-governance and their cultural identity and autonomy [67]. Further, the closure of Karen schools caused by restrictions on access to education has forced children into Thai schools where they may face language and cultural barriers. These issues were exacerbated by the 1991 designation of the TYNWS and the adjacent Huai Kha Khaeng WS as a UNESCO World Heritage Site, further exemplifying the critical tensions between Indigenous rights and the preservation of natural and cultural heritage [65–67].

According to the nomination document, the Karen communities living within the sanctuary are considered a potential hazard to its preservation. Consequently, preparations are underway to relocate them in the near future [66,67]. This situation raises concerns regarding the criteria followed by international organizations such as UNESCO in the designation of cultural and natural sites given their mission is to protect Indigenous rights as well as preserve natural and cultural diversity. Questions arise on whether, in their endeavors to safeguard natural environments and biodiversity, these institutions may occasionally fail to protect or even consider the very communities that have inhabited these territories for generations. Situations like this highlight the urgent need for dialogue and cooperation among Indigenous communities, local authorities, and international institutions to find a balance between the protection of IP rights and biodiversity conservation. To this end, decision-making processes must consider the perspectives, traditions, customs, and historical connections of the affected Indigenous communities with the land.

Although certain recent advancements have been made, such as the acknowledgment of the Karen people as an Indigenous group in 2019, these individual measures are clearly insufficient to solve the numerous obstacles they continue to encounter today. To protect the rights and dignity of the Karen people, greater efforts are required to ensure equitable land rights, enhanced access to education and health care, and the elimination of violence and discrimination. The preservation of these rights is essential for the Karen people to live with the dignity and respect they deserve within the legal framework of Thailand.

## 4. Case Study 2: Indigenous People of Yakutia Region (Sakha Republic, Russia)

### 4.1. Historical Background

Contrary to Thailand, where Indigenous people are clearly differentiated from the rest of the population, the definition of Indigenous peoples in Russia is linked to population size and natural resource use practices [68]. Thus, although about 200 different nations live on the territory of the Russian Federation, the majority of ethnic Russian origin (Slavic people), only 47 nations are considered by the state to be Indigenous people as officially documented in the register of Indigenous small-numbered peoples (several of which are represented within the Republic of Sakha; Figure 1b). These are defined as Indigenous nations with populations smaller than 50,000 people and distinct traditional cultures and livelihoods [69,70]. Of the 47 officially recognized Indigenous nations, 40 groups are geographically classified by Russian legislation as 'Indigenous peoples of the North, Siberia, and the Far East' often engaging in natural resource use practices such as reindeer herding, hunting, and fishing [69–71]. In addition to the officially recognized Indigenous minority groups, several other ethnic groups live across the Sakha Republic such as the Russian Old-Settlers or Old-Timers (Russkoustinians), descendants of the first European colonists who settled the Arctic shores of Eastern Siberia during the 16th century, or the Yakuts (Sakha people), who represent the most populous ethnic group in the Republic of Sakha [71].

A wide variety of Indigenous settlements, each possessing a distinct cultural identity and heritage, populate the vast territory of the Republic of Sakha [72]. Through their long-established traditions that have sustained their communities for generations, these settlements make substantial contributions to the economy and cultural diversity of Yakutia, acting as custodians of its rich natural environment and cultural heritage. Traditional activities such as reindeer herding, fishing, or hunting represent for these IPs not just their livelihoods but a way of life that provides a profound bond to their lands and traditions, with generations passing down traditional skills and wisdom [73]. Their deep respect for the land and its resources is evident in their sustainable land management practices. For example, IP in the Republic of Sakha has traditionally used controlled fire to burn accumulated combustible material as a fire-fighting land management tool for centuries [74] However, the ban on this practice has contributed to the numerous forest fires that have ravaged the Republic over recent decades [74]. Their dedication to the preservation of their cultural heritage is of equal importance as evidenced by the distinctive ways in which their languages, folklore, and traditions are transmitted in the form of tales,

songs, and rituals from elders to younger generations. In doing so, these communities become living repositories of the region's abundant cultural heritage and traditions in the modern era. Therefore, the IP of the Republic of Sakha exemplifies the enduring synergy between humanity and the environment, as well as the resilience of their diversified Indigenous populations.

Prominent ethnic groups in Yakutia include the Chukchi, Dolgan, the Evenk, and the Russian Old-Timers. The Sakha people also called the Yakut, are one of the major groups in eastern Siberia [71]. Of Turkic origin, they expanded from their initially limited area on the middle Lena River in the 17th century to their current presence over much of Yakutia. Despite the harsh climatic conditions that make their livestock dependent on shelter and feeding for a large part of the year, the Sakha have skillfully clung to an economy based on the raising of cattle and horse breeding [69]. Dairy products and meat take prominent places in their diet while fishing in the abundant rivers and lakes represents their second most important traditional activity [72]. Similarly, meat from reindeer herds, wild game, fish, or marine mammals represents the base of the TFS for different Indigenous groups of the Far North such as Nenets, Dolgan, Evenk, and Chukchi. Meat is consumed raw, from freshly killed or merely wounded animals, as well as cooked (boiled or grilled) and preserved using traditional techniques such as fermentation, dry-curing, or, like the Yukaghir, stored in the frozen ground [72]. The stroganina is a traditional dish from northern Siberia comprising long, thin slices of frozen raw meat or fish [72]. Meat-based diets are supplemented with edible herbs and plants, berries, and other types of accessory foods [72]. For example, the Yukaghir consumes different edible plants, like wild onion, and day lily roots, as well as berries and mushrooms.

The Sakha cuisine is set aside in that it is influenced by elements of both Arctic and Mongolian cuisines [72]. It relies heavily on horse meat as the Sakha are expert horse breeders. Dairy products also form an important part of their diets as they also raise cattle. The Kymys, for example, is a very popular drink made from fermented mare's milk. Fish is also a prominent product of their diets, especially Siberian sturgeon, broad and northern whitefish, Arctic cisco, muksun, and grayling [72]. However, Russian colonization of Yakutia in the 1600s gradually changed the traditional diet structure of all these Indigenous groups, especially the Sakha, introducing products like flour, grains, salt, sugar, tea, and alcoholic beverages and borrowing culinary practices, especially soups and mushrooms consumption [72].

### 4.2. Legal and Regulatory Frameworks in the Indigenous Settlements of Yakutia (Russia)

Unlike the Karen Indigenous communities in Thailand, where access to forests and natural resources are regulated by the Royal Thai Government, land accessibility and the use of natural resources by the Yakutian Indigenous communities are governed by formal and informal rules based on family, kinship, or tribal proximity that have been developed over centuries in the context of political and economic changes in Russia [73]. Formal rules regarding access to and development of natural resources only began to be introduced in the 1920s during the Soviet industrialization and collectivization of Yakutia, which was accompanied by a new administrative-territorial division of Yakutia. The state policies on collectivization during the Soviet period in Russia attempted to unite small rural households, including its Indigenous peoples, into collective farms and build settlements around those farms [74]. During this time, the rights to use natural resources for fishing, hunting, gathering, reindeer husbandry, or other traditional IP activities became formally regulated through the institutionalization of collective farms [73,74].

The subsequent breakdown of the Soviet Union brought legal and economic reforms that led to the dissolution of many of these collective farms, although some remnants of these organizations still exist today and lead economic activities. However, the Indigenous peoples were often given a chance to form smaller units or organizations to lead traditional natural resource practices. Along with the law on traditional natural resource use of Indigenous peoples of the North, new forms of organizations have emerged, such as

tribal communes [74]. A tribal commune is a small organization that allows members of the same family or kin to engage in 'traditional natural resource use' practices such as reindeer herding, fishing, and hunting. These tribal communes started to gain legal rights to their traditional lands under the regulation on 'territories for traditional natural resource use' [75,76]. As of 2020, Indigenous peoples of the North of Yakutia have registered 62 such territories across 21 of the 33 districts of Yakutia [75]. These territories are not owned by the tribal communes (or other forms of organizations of Indigenous peoples such as limited companies, etc.) but are awarded by the federal state for specific use. Overall, the legislation on land, forests, fisheries, and hunting is basically developed at the federal level. Individual regions in Russia (such as the Republic of Sakha) have limited capacity to introduce changes to the implementation of these federal laws.

At a national level, the Russian Federation has adopted a series of successive national laws that represent the current legal framework on cultural, territorial and political rights of Indigenous communities (the Federal Law on the Guarantees of the Rights of the Indigenous Small-Numbered Peoples of the Russian Federation adopted of 1999; the Federal Law on General Principles of Organization of Obshchina of Small-Numbered Indigenous Peoples of the North, Siberia and the Far East of the Russian Federation of 2000; and the Federal Law on Territories of Traditional Nature Use of the Small-Numbered Indigenous Peoples of the North, Siberia and the Far East of the Russian Federation of 2001). Whereas these laws were enacted to guarantee IP's rights on their traditional land, to participate in decisions about resource exploitation and conservation, and to have access to fair compensation for eventual damages from industrial and economic development, the reality is that recent amendments to all these laws have made the actual implementation of these rights virtually impossible and clearly seek to legally disempower and exclude Indigenous peoples from the management of their ancestral lands [77]. Nonetheless, some scholars have also noted that this complex structure has created resistance to the decisions made at the federal level in some regions, including the Republic of Sakha. This makes the protection of the rights of Indigenous peoples in the Republic relatively stronger than in other regions of Russia.

### 4.3. Implications of Legal and Regulatory Frameworks on the ISES and TFS of Indigenous Communities in Yakutia (Russia)

The legal and regulatory frameworks in Yakutia wield considerable influence over the Indigenous communities inhabiting this region, resulting in multifaceted impacts that significantly shape their way of life [78]. Access to land and natural resources, essential for the traditional livelihoods of these Indigenous groups, has become increasingly challenging due to state or private control over many of these resources [78]. This has drastically limited the ability of Indigenous communities to secure the resources vital for their survival and prosperity. Moreover, the legal and regulatory landscape often prioritizes large-scale economic development projects at the expense of the traditions and cultures of the Indigenous communities [78]. The absence of adequate protection for the cultural rights of Indigenous groups is a significant concern, such as the widespread lack of the basic right to free, prior, and informed consent before development projects are undertaken on their ancestral lands [79].

Examples of these impacts are evident in several notable cases. For instance, the construction of the Sakhalin-Khabarovsk-Vladivostok oil pipeline has severely impacted the livelihoods of the Evenki and Yakut Indigenous communities in Yakutia [79,80]. This massive infrastructure project has disrupted the traditional migration patterns of reindeer and other animals, which represent the base of their TFS. The construction has additionally created severe environmental pollution problems and damaged different cultural heritage sites of immense significance for their identity and spirituality [80,81]. Similarly, the recent development of the Yakutia gas fields in the remote Arctic regions of the Republic has resulted in the forced displacement of Indigenous communities from their ancestral lands into towns and cities, where they often grapple with difficulties in finding employment and preserving their cultural identity [82]. Finally, the complexity and dysfunctionality of

the interdepartmental structure imposed by the state regulation of hunting and the lack of legislative support for enforcing priority rights of small-numbered Indigenous peoples on the use of hunting and fishing resources contributes to the increasing difficulties faced by Indigenous communities in preserving their traditional subsistence practices [82,83]. All these laws have exacerbated conflicts between Indigenous communities and the government, creating tensions that challenge the cultural and economic equilibrium.

One of the main difficulties for Indigenous peoples to engage in natural resource use practices relates to the complexity of the relationship between the federal state and regional administrative, executive, and legislative powers. There is a division of ownership, control, and governance for land, forests, and water bodies, including the administration of rights and environmental protection [84]. For instance, private land ownership is only possible within the boundaries of municipalities such as villages and towns, while almost all forested land in Russia, where IPs conduct their traditional natural resource activities, belongs to the federal state. Thus, the importance of federal administration in regional natural resource management is of paramount importance, often overpowering the interests and concerns of regional populations and, consequently, affecting the routines and natural resource practices of Indigenous and non-Indigenous populations. Changes in Russian legislation and accompanying regulations and procedures for natural resource use activities in forests and water bodies such as hunting, fishing, and plant gathering are often enforced without specific regard to the interests of populations and consideration of regional specifics of the natural environment [84].

Despite these adversities, the Indigenous communities of Yakutia have exhibited remarkable resilience and determination in protecting their rights and heritage. They continue to engage in efforts to preserve their culture and secure sustainable economic development projects that benefit their entire community. These endeavors underscore the enduring spirit and commitment of these communities to maintain their unique identities and way of life in the face of formidable challenges.

## 5. Examining the Political-Ecological Theoretical Framework (PETF) as Applied in Our Study

In this section, we revisit and expand on the three main dimensions of PETF within the context of the ISES and TFS for our comparative case study:

- Power dynamics and Indigenous rights: Both the Karen and Yakutia Indigenous communities have experienced historical inequalities in power structures. Indigenous communities in their endeavors to assert their rights to their territories, resources, and traditional knowledge may be empowered or disenfranchised by these power dynamics. The consequences of colonialism and policies for the Yakutia Indigenous Groups during the Soviet era have contributed to the formation of a milieu wherein Indigenous communities frequently encounter obstacles when attempting to assert their rights. Historically, the consolidation of authority has restricted the capacity of these individuals to exert influence over determinations that pertain to their territories and valuable resources. Power imbalances have also developed between the Indigenous people of this region and the government because of the growing pressure for resource extraction, which is frequently motivated by external economic interests without free, prior, and informed consent. As economic agendas prioritize extraction over the preservation of Indigenous territories and traditional practices, the rights of the Yakutia IPs have been often compromised. This has resulted in environmental degradation and posed increasing challenges to the sustainability of their SES-TFS. Although legal frameworks in Russia acknowledge and protect Indigenous rights, their implementation has been hampered by the complex interdepartmental principle of regulatory structure. The recognition of Indigenous land rights and the right to practice traditional livelihoods is crucial, but gaps in enforcement and disparities in legal interpretation often impede the full realization of these rights. On the other hand, the Karen IPs have historically struggled with challenges related to land rights, facing

increasing difficulties in securing and maintaining their ancestral lands, which have led to profound power imbalances. As discussed before, state policies have prioritized other interests resulting in the frequent systematic marginalization and violation of Karen rights. Furthermore, the presence and participation of military forces in regions inhabited by Karen communities for the implementation of state policies, including forced relocation and eviction, have created power dynamics that adversely affect Indigenous rights. Armed conflicts and militarization pose threats to the security and well-being of the Karen people, limiting their participation in decision-making processes related to land use, resource management, and the preservation of their cultural heritage. Lastly, state conservation policies aiming to protect natural resources have come into direct conflict with the Karen communities living in the region and their traditional ways of living. Forest conservation measures and the creation of wildlife sanctuaries and protected areas, as discussed before, have severely restricted their access to traditional lands and resources and their ability to sustain their extraordinarily diverse traditional food systems. It can therefore be pointed out that negotiating a balance between conservation goals and Indigenous rights remains a challenge to be addressed. In conclusion, power dynamics can either empower or disempower Indigenous communities in asserting their rights over their lands, resources, and traditional knowledge. In our two case studies, there is clear evidence that power dynamics have disempowered Indigenous communities in asserting their rights over their lands, resources, and the sustainability of their SES-TFS.

- Sociopolitical contexts and environmental implications: The sociopolitical landscape of the Karen communities is characterized by a history of continuous land rights disputes, military interventions, confrontations, and obstacles presented by conservation policies. This historical marginalization of the Karen people has been exacerbated by their limited political representation, which hinders their capacity to champion Indigenous rights and safeguard their cultural heritage. In contrast, the sociopolitical dynamics of Yakutia Indigenous communities have been profoundly influenced by the centralization of power structures, resource extraction pressures, and the repercussions of Soviet-era policies. The limited control over land use and historical inequalities underscores the need for inclusive policies that respect Indigenous rights and cultural heritage in both settings. Limited political representation affects the ability of IPs to advocate for their rights at the governmental level in both case studies where decisions to protect and govern ISES follow a clear top-down structure that is enforced on the Indigenous communities. The lack of participatory and inclusive decision-making processes has exacerbated these power imbalances, hindering the effective protection of the ISES-ITFS. Efforts to preserve the socio-cultural identity and traditional knowledge among the Karen and the Yakutia Indigenous people face challenges due to multiple external pressures and power imbalances that need to be understood and corrected to develop fairer and more effective legal frameworks that protect their rights and ISES-ITFS.

- Unequal resource distribution and historical marginalization: We have seen how the Karen and Yakutia Indigenous peoples have been subjected to historical marginalization and the inequitable distribution of resources in their lands. Land dispossession, displacement caused by conflict, and inadequate political representation have all played a role in the Karen people's persistent difficulties in maintaining and preserving their ITFS. On the other hand, historic legacies in Yakutia, including centralized power structures, resource extraction that prioritized economic interests, and the repercussions of Soviet-era policies, have restricted IP control over the access and use of Yakutia's abundant natural resources. To rectify these past inequities, it is imperative to implement comprehensive legal reforms that properly acknowledge and protect the rights of IPs over natural resources in their lands and foster inclusive decision-making processes that empower these communities.

## 6. Discussion

Our combined analysis of existing literature reviews on the topic, national legal and regulatory laws governing the Indigenous territories in our study communities, and a comprehensive analysis of international Indigenous rights reports clearly exemplify the connection of the Karen people with their lands through a rich history of traditions deeply intertwined with their natural environment. However, the introduction of legal and regulatory frameworks, particularly during the colonial and post-colonial eras, has resulted in the continuous erosion of their cultural identity and access to their TFS. The persistent assimilation policies enforced by the Royal Thai government have targeted the suppression of the Karen language, customs, and religious practices. These policies, together with land-related conservation regulations, have continuously disrupted their traditional subsistence activities and impacted their TFS. Today, the Karen people continue to face displacement, loss of livelihoods, and social marginalization, resulting in the instigation of opposition and resentment amongst these communities. Karen traditional means of subsistence, including gathering, farming, hunting, and foraging, have become increasingly difficult to practice due to the loss of legal ownership of their native lands. The quality of life for numerous Karen residing in remote, undeveloped areas is also negatively impacted by their inability to access proper education and health care. The frequent violence and discrimination employed by both the military and the government to impose the state law have often culminated in forced displacement and even torture and extrajudicial executions for the opposition.

Nonetheless, some recent positive developments have also occurred, like the official recognition of the Karen in Thailand as Indigenous people. However, these individual advancements are still clearly insufficient. A much more ambitious holistic approach is required by the Thai government to ensure that the rights and dignity of the Karen people are properly recognized and respected. At a minimum, these initiatives should guarantee fair and just land rights, enhance opportunities for education and health care, and confront the persistent challenges of violence and discrimination that impact these communities. Therefore, the Karen Indigenous people of Thailand serve as a case study that demonstrates the severe consequences that Indigenous communities worldwide can endure due to inadequately designed legal and regulatory structures. This reality has been recognized by other scholars who have investigated the Karen Indigenous communities in Thailand [61–63].

Despite all these misfortunes and challenges to their rights and existence, the Karen people have repeatedly demonstrated exceptional fortitude and a resolution to find alternative channels to exert pressure on the government and institutions to create the conditions that may eventually change the existing legal and regulatory status quo. They have adopted a multifaceted strategy based on advocacy, legal action, community development, and international cooperation to assert their rights [63]. Driven by their desire for fairness and impartiality, they have utilized advocacy and campaigning as effective strategies to generate national and international awareness of their hardships and advance their entitlements. Legal contests represent an additional crucial aspect of the Karen people's endeavors to deconstruct discriminatory policies and laws. The Thai government has been indicted in legal proceedings attempting to rectify the infringements upon their cultural and land rights [65,66]. These legal actions provide a tool for the Karen people to challenge the existing legal structure that has sustained their marginalization and injustice. Additionally, the Karen people place considerable importance on community development to enhance their overall standard of living. Their community-building endeavors are supported by initiatives such as the formation of agricultural cooperatives, healthcare institutions, and schools [65–67]. These enterprises not only promote self-sufficiency but also facilitate the conservation of their cultural legacy, ultimately elevating the standard of living in their respective communities.

In addition, recognizing the current global support for Indigenous rights, the Karen people have strategically established international networks to support their cause and

used the mainstream media and social platforms for exchanging information and collaborating with human rights organizations and other Indigenous communities around the world. Certain organizations have assumed a vital role in recent years in advocating for the rights and interests of the Karen people in Thailand. For example, the Karen Human Rights Group (KHRG) is a non-governmental organization whose mission is to advocate for justice on both the national and international levels while documenting human rights violations against the Karen people. Similarly, the Karen Women's Organization is dedicated to advocating for the rights of Karen women and girls and providing support to access essential services like health care and education. Finally, the Karen National Union is a political organization that strives to protect the Karen's right to self-determination. To improve the welfare of Karen communities, the Karen National Union has also established many social services, including healthcare clinics and schools. Therefore, the resolute dedication to advocacy and progress of the Karen people has resulted in substantial advancements in their endeavor to assert their rights providing hope for a better, fairer future for future generations. Ultimately, the Karen people's ordeal reflects the enduring spirit of Indigenous communities worldwide as they seek justice, equality, and the preservation of their rich cultural heritage. These examples show how IPs can influence existing legal and regulatory frameworks and highlight the importance of their actions to trigger change including taking advantage of new opportunities, such as the growing role of young generations in empowering Indigenous rights and advocacy using social media and digital technologies.

In contrast, the IP of Yakutia in Russia faces a very different set of legal and regulatory challenges. Their traditional access rights to their land and its natural resources are increasingly curtailed by increasing state or private control. Historically, the access to, and use of, land and natural resources has been regulated by informal rules based on family, kinship, or tribal proximity. Formal rules were introduced during the Soviet period when the state aimed to consolidate control over these resources by organizing collective farms. The dissolution of the Soviet Union subsequently allowed Indigenous communities to form smaller units or organizations to continue their traditional natural resource practices. However, as previously discussed, current legal and regulatory frameworks in the Sakha Republic tend to prioritize large-scale economic development projects over Indigenous rights and traditions. These have significantly affected the sustainability of ISES-TFS through a series of impacts, including the disruption of migration patterns of reindeer and other wild animals, increased environmental pollution, and damage to culturally important sites. Similarly, land transformation from extractive industries, such as mining or gas fields, has resulted in forced relocations and widespread environmental damage. Further, Indigenous communities in Yakutia have often lacked the right to free, prior, and informed consent before such development projects are initiated on their lands, undermining their cultural autonomy and their traditional ways of life as earlier discussed. Despite these challenges, the Indigenous communities of Yakutia remain determined to protect their rights and heritage, engaging in efforts to preserve their culture and secure sustainable economic development projects.

These two case studies illustrate the complexity of the relationship between the state and the regional administrative, executive, and legislative powers not only in Russia and Thailand but across the globe. They clearly underscore the importance of understanding and addressing the impacts of legal and regulatory frameworks on Indigenous communities as the outcomes range from adversity to resilience shaping their lives in profound ways. In a broad sense, case-study approaches are particularly useful when the objective is to obtain an in-depth understanding of complex issues in real-life settings. In our case, the case study approach provides several clear advantages. Firstly, the study offers cross-cultural insights into how legal and regulatory policy frameworks are impacting the resilience and sustainability of the ISES and ITFS in the study communities. Similarly, by analyzing the parallels and variations in regulatory and legal frameworks among these Indigenous groups and their impacts, our study bridges the existing knowledge gap and guides policy-

making processes by emphasizing shared challenges, vulnerabilities, and experiences of Indigenous communities not only in our study communities but across the globe.

On the other hand, case study approaches are sometimes criticized for lacking sufficient scientific rigor, posing difficulties in extrapolating findings, and being prone to researcher bias. In our case, we believe that the proposed methodology for assessing the impacts of legal and regulatory frameworks on ISES-ITFS is broad in scope and inclusive in concept to be applicable elsewhere, while the results from our case studies should be applicable to many other regions where IPs are confronting similar situations regarding the impacts of existing legal and regulatory frameworks on their socioecological and food systems. On the other hand, we try to be as rigorous as possible with our analysis by doing an exhaustive search of data and information that covered as broad range of sources as possible, including scientific papers, literature reviews, relevant legal documents, and international Indigenous rights reports. Nonetheless, our approach encountered limitations, especially in contextual variations between the two case studies, issues with data availability and ethical concerns on this very sensitive topic, as well as translation obstacles that required careful interpretation and awareness to maintain the accuracy and dependability of research results while honoring Indigenous viewpoints and rights.

## 7. Policy, Advocacy, and Recommendations

Our study has significant implications across various areas (policy, application, practice, and theory) within the Political Ecology Framework (Figure 1) for sustainable ITFS that addresses power issues associated with sovereignty over Indigenous land and natural resources, recognizing Indigenous rights, and identifying the role of inclusivity and justice in governance and management. Here, we outline the policy implications, building upon the Political Ecological Theoretical Framework, for the resilience and sustainability of the ITFS for our comparative case studies.

*Implications of This Comparative Research Study*

- Policy implications: This comparative analysis underscores the urgent need for policy reforms at both national and international levels. Our study highlights the urgency of acknowledging Indigenous land rights, advocating for sustainable resource management, and tackling the issue of climate resilience through the active participation of all stakeholders including IP. Governments should consider implementing or amending policies that respect the rights, culture, and traditional knowledge of Indigenous communities. We recommend that governmental bodies worldwide should promote the formulation of policies that seek to incorporate Indigenous practices into more comprehensive strategies for conservation and sustainability purposes as these have proven to play an important role in the sustainability and resilience of ISES and ITFS.
- Application in Indigenous communities: In addition, our research identifies practical recommendations for navigating the legal and regulatory framework by fostering sustainability and resilience among Indigenous communities. The significance of Indigenous and community-led initiatives working in tandem with other stakeholders to attain climate resilience and conservation objectives is underscored in our two case studies. By identifying and confronting the challenges and opportunities that arise from navigating legal and regulatory frameworks within the ISES, we believe that new knowledge on how best to manage ISES and ITFS will be provided.
- Practice and cultural preservation: An essential component of this research pertains to the conservation of Indigenous cultures. Indigenous communities worldwide have, for generations, amassed knowledge and distinctive practices that are indispensable for the management of ecosystems [19]. Therefore, collaborative initiatives based on the principles of co-management and inclusive participation pairing Indigenous ecological knowledge and traditional practices with scientific knowledge and conventional management are a promising way to advance toward the long-term sustainability and resilience of ISES-ITFS amid the environmental and climate change crises.

- Theoretical contributions: Our research makes a theoretical contribution to the discourse surrounding the intricacies between ISES and the legal and regulatory frameworks that can have a profound impact on Indigenous communities and their quest for the sustainability and resilience of ITFS. Our analysis further contributes to the discourse on environmental justice, the correlation between culture and conservation, and the significance of Indigenous knowledge in the context of sustainability.
- International collaboration: In addition, this research highlights the importance of international cooperation, the exchange of information, and assistance for Indigenous communities. It emphasizes the significance of collaborative initiatives that foster international human rights frameworks and Indigenous networks. Our study encourages international collaborations that may aid Indigenous communities in navigating legal and regulatory frameworks that are poorly crafted by bringing these concerns to light.

In summary, our comparative case study offers a comprehensive understanding of the challenges faced by Indigenous communities in two distinct regions. The implications of this study span from practical applications to theoretical contributions (Table 1). Policy changes, culturally sensitive practices, and international collaboration are crucial in promoting the climate resilience and sustainability of ISES-ITFS.

**Table 1.** Identified areas and ways of action towards a collaborative approach in enacting policies governing the sustainability and resilience of the ISES and ITFS.

| Areas of Action | Ways of Action |
| --- | --- |
| Indigenous knowledge and perspectives | We recognize that IPs possess a wealth of place-based ecological knowledge accumulated over generations that is instrumental to the resilience and long-term sustainability of ISES and ITFS. Therefore, it is imperative to promote initiatives seeking to integrate this knowledge (TEK) into the formulation of legal and regulatory frameworks affecting the sustainability of ISES-ITFS. |
| Legal and regulatory frameworks | We understand that legal and regulatory frameworks can provide a strong foundation for conservation by setting clear rules and regulations that protect biodiversity and natural resources. However, Indigenous peoples should be involved in the development of these frameworks to ensure that they are compatible with their traditions and subsistence activities. |
| Institutional arrangements | We reiterate that the development and establishment of effective institutional structures is vital for the effective functioning of legal and regulatory frameworks, especially those pertaining to conservation and the management of ISES and ITFS. To this end, it is crucial that Indigenous communities actively participate in the conceptualization and execution of these legal and regulatory instruments to guarantee that they address their needs and concerns. |
| Capacity building, Indigenous knowledge, and research | Because Indigenous peoples often lack the resources and capacity to participate in the development and implementation of legal and regulatory instruments for conservation., we assert that IP must be given the necessary means and support to continue exerting their rights over their ancestral territories and natural capital as a way for building resilience and adaptive capacity. This should also include the requirement for all institutions and researchers to honor Indigenous sovereignty and rights over their traditional knowledge and data governance. Similarly, we believe that facilitating IP's access to funding programs and the benefits resulting from the research conducted in their lands should be also become mandatory for governments as well as academic and research institutions. |
| Monitoring and evaluation | Finally, we reiterate the urgency of regularly monitoring and evaluating legal and regulatory structures that are developed to promote the sustainability, resilience, and sustainability of ISES-ITFS. Such continuous evaluation should allow identifying and correcting deficiencies while reinforcing the effective components of legal and regulatory frameworks. We believe this is necessary to ensure that these instruments remain effective and continue to address the needs of Indigenous people by reinforcing elements and correcting deficiencies. |

### 8. Conclusions

This study offers valuable insights into the intricacies of legal and regulatory frameworks and their potential impacts on the sustainability and resilience of the ISES and ITFS. The analysis of our two case studies highlights the importance of recognizing the diversity of ISES and the need for context-specific policies that respect Indigenous knowledge and rights. While Indigenous communities continue to face unique challenges, common threads in their experiences underscore the need for comprehensive legal and regulatory frameworks that prioritize Indigenous self-determination, cultural preservation, and rights over land and resources. These frameworks should recognize the value of Indigenous traditional and ecological knowledge and be inclusive and participatory to reflect Indigenous values, knowledge, and aspirations. To navigate the path towards climate resilience and sustainability within ISES, governments and international organizations must recognize the agency of Indigenous peoples and engage them as active partners in climate adaptation and mitigation efforts. This recognition should extend beyond cosmetic gestures and individual concessions to meaningful collaboration and decision-making power for Indigenous communities.

In conclusion, our study serves as a timely reminder that safeguarding Indigenous socioecological systems, their rights, and their unique knowledge is not only a matter of justice but also a crucial strategy for the attainment of the global climate and sustainability agendas. We believe that future research and policy development in the topic of regulatory frameworks for the sustainability and resilience of the ISES should focus on the following research areas:

- Indigenous legal frameworks: Research should focus on the development and effectiveness of Indigenous legal frameworks within the context of national legal systems by exploring how they can better align with Indigenous values and customs to protect ISES.
- Community participation and empowerment: Investigate strategies for enhancing Indigenous engagement in the development and implementation of legal and regulatory frameworks, recognizing the importance of meaningful consultation and self-determination.
- Transboundary collaboration: Investigate mechanisms that facilitate efficient collaboration on legal issues involving Indigenous territories that transcend national boundaries to promote transboundary cooperation that protects shared resources.
- Cultural preservation and knowledge integration: Examine how legal and regulatory frameworks can better incorporate and protect Indigenous traditional ecological knowledge and cultural practices by ensuring they are integral components of legal policies and regulatory practices that affect ISES.

**Author Contributions:** Conceptualization: S.C.S. and J.G.M.; methodology and analysis: S.C.S.; data compilation: S.C.S., V.P., T.G. and N.Y.; writing of the original draft: S.C.S., T.G. and N.Y.; supervision: J.G.M. All authors have read and agreed to the published version of the manuscript.

**Funding:** This work was supported by the Japanese Science and Technology Agency (JST SICORP Grant Number JPMJSC20E5) and the Russian Foundation for Basic Research (RFBR project number 21-55-70104) as part of the East Asia Science and Innovation Area Joint Research Program (e-ASIA JRP) for the Climate Change Impact on Natural and Human Systems.

**Informed Consent Statement:** This study did not involve any human subjects; as such, no informed consent was required.

**Data Availability Statement:** No data was used or generated by this article.

**Conflicts of Interest:** The authors declare no conflicts of interest.

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
