# Peer review of "Navigating Legal and Regulatory Frameworks to Achieve the Resilience and Sustainability of Indigenous Socioecological Systems"

_resources, doi:10.3390/resources13040056_

Round 1

Reviewer 1 Report

Comments and Suggestions for Authors

Through its use of a comparative case study approach, this paper provides important insights into the effects of regulatory and legal frameworks on Indigenous Socioecological Systems and Indigenous Traditional Food Systems. The two case studies are well developed, providing examples of ways in which existing regulatory and legal structures are having a sustained negative impact on Indigenous peoples and communities. The paper also provides insights into resilience and sustainability on the part of Indigenous peoples, and ways in which they are engaging with and contesting these legal and regulatory regimes. The case studies provide support for the explanatory capacity of the proposed PETF.

Overall, the paper engages with a series of critical issues, and provides new theoretical and empirical insights that will be of interest to other researchers. The paper is well written and structured.

The following are comments for the authors to consider:

Theoretical and conceptual points

Power is a central concept in the paper, including for the Political Ecological Theoretical Framework. Given that power is a contested concept, it would be beneficial if the authors provide the definition of power that they are using in the paper, and discuss why they are using that particular definition. Related to this, a clear definition of the concept of ‘power dynamics’ is also needed, as providing an important foundation for the contribution of the paper.

In the discussion of the PETF model (lines 242ff), it is not clear if this model is being adopted from other researchers, or whether it is a model that the authors are proposing.  This point needs clarification.  That is, when the authors say they are adopting this framework, does this mean they are developing it, or they are making use of a framework developed by other researchers. 

In the discussion of the PETF, it is stated that the framework facilitates examination of power dynamics, see line 792ff. While power emerges from the case studies as a critical issue, the argument as to how the framework enables the analysis of power would benefit from further development. For example, could the concepts of power and power dynamics be represented directly in the framework figure?

In the PETF, the importance of social, political, environmental and economic factors is identified appropriately. It is not clear at present, however, as to why social and political are grouped together, and environmental and economic are grouped together. Are these groupings intended to suggest something about the relationships between these factors, which then influences the legal and regulatory framework? 

Does the PETF model have a way of acknowledging that Indigenous people may be able to influence the legal and regulatory framework through their actions and agency?

Methodological points

The comparative case study approach is appropriate for this study. However, the key dimensions of a comparative case study approach could be discussed in more detail. In this regard, in the methodology section (1.4), it would also be helpful for the authors to provide further discussion on the benefits and limitations of case study approaches. 

In setting up the case studies, the authors refer to a range of document sources (lines 221-225). It would be helpful to provide more detail on these documents.  For example, how many documents in each category have been used in the analysis. And if the numbers of documents are manageable, is it feasible, perhaps in an appendix, to provide a table that lists the documents being analysed? This would help the reader to understand the data being used as the basis for the paper. And following this, how have the documents been analysed?

The methodology mentions the RISE project of the authors. It would be helpful to provide more discussion on the focus of the RISE project, if data from the RISE project is being used in the paper. In addition, it is noted that on-site observations from the RISE project inform the paper (e.g. line 875). The informed consent statement at the end of the paper indicates, however, that the study did not involve human subjects. It is crucial that the authors clarify whether the observations informing the paper involved human subjects.  This is important in all research, and in particular in the context of research with Indigenous peoples, where Indigenous data sovereignty and data governance is a critical issue.

The ISET model

Section 5, and following, moves from a consideration of the case studies and the value of the PETF framework, to a discussion of policy, advocacy and recommendations. While there is an important need for researchers to engage in such discussion, at this stage, this section (lines 1001 to 1090) needs further consideration and development. In particular, in this section the authors propose an Indigenous Socio-Ecological Theoretical Model, building on the PETF. At the centre of the ISET model is co-management. While co-management is an important approach, at present there are four key challenges with how this approach is being used in the paper.   

First, it is presented in the context of ‘we believe’ (line 1025; 1036). That is, co-management is being presented as a preferred approach based on the beliefs of the authors. To be supportable, the discussion of the approach needs to be grounded in evidence.  Earlier parts of the paper contained some discussion of co-management. This could be returned to more fully in this section. This is of particular importance in the context of the discussion in lines 1031 to 1034, where the authors discuss collaborative governance structures. To be persuasive, this needs further development.

Second, the discussion of co-management (lines 1017 to 1089) proceeds without any references to existing research. Without references, it is not clear what the claims being made are based on. 

Third, as currently presented, it is not clear how the ISET model is built on the PETF model (line 1017).  The connections between the two models need more discussion.

Fourth, the ISET model itself needs further discussion, to set out the proposed relationships and outcomes, and what they are based on.

Overall, the ISET model has potential to be an important contribution, both theoretically and in terms of practice. At present, however, the discussion needs considerable further development. As a suggestion for the authors to consider, the discussion in section 5 on the ISET model could be the basis for a separate paper. One option would be to omit this part from the current paper. The authors could keep the focus of this paper on the comparative case study and the PETF model. This would constitute an important contribution in itself. The ISET model could then be the basis for a separate paper. If the authors wish to keep the discussion of the ISET model and the argument for co-management in the current paper, the development and justification of the model requires further work.

Other points for consideration

Lines 369ff and 389ff describe some success stories around collaborations between Indigenous peoples and national governments, represented in legislation.  While these are important examples, it is also important not to overstate the progress that has been made. In the Australian context, for example, while the Mabo decision and the 1993 Act are significant, it may be going too far to say that an outcome has been “a harmonious and mutually beneficial relationship between governments and Indigenous peoples.” At the least, the authors should note that ongoing challenges remain, as Indigenous people in Australia continue to experience racism and significant forms of inequality across all areas of Australian society.  Such challenges are related to ongoing power struggles in Australia, as non-Indigenous Australians seek to maintain their dominant position within Australian society. See, for example, the work of Aileen Moreton-Robinson; Marcia Langton; and Emma Lee.

A critical issue for the paper involves the engagement of researchers with Indigenous knowledge. In the ‘areas of action and ways of action’ table, another area of action could be included which is focused on Indigenous knowledge and research, identifying the need for all researchers to consider issues such as Indigenous data sovereignty and data governance.

Through the paper, the authors use the acronym IP for Indigenous people. I recommend spelling out Indigenous people each time. IP may be confused with intellectual property, for which it is a common acronym.

Line 321, the sub-heading might be more accurately presented as: Traditional Ecological Knowledge (TEK) protection. This heading then links more directly to the discussion that follows.

Through the paper, the authors need to ensure they are providing full referencing for claims being made.  As just one example, the paragraph from line 914 to 934 contains a number of examples of actions being taken by the Karen people. But no references are provided, so the reader does not know what the examples provided are based on in terms of data. The same issue continues in a number of the following paragraphs.

Comments on the Quality of English Language

The paper is well written and the quality of English language is appropriate.

Reviewer 2 Report

Comments and Suggestions for Authors

I have reviewed the manuscript titled 'Navigating Legal and Regulatory Frameworks to Achieve the Climate Resilience and Sustainability of Indigenous Socioecological Systems (ISES).' This study aims to primarily analyze the impact of policy and legal frameworks on ISESs, discerning ways in which favorable policies can effectively support Indigenous rights, wisdom, and practices, while also investigating the harmful effects of poorly designed frameworks rooted in historical marginalization. The study addresses an original topic, although this isn't clearly articulated within the paper. There are some shortcomings in the study that need addressing. After these shortcomings are rectified or satisfactory explanations are provided by the authors in response to my critiques, a reevaluation could be warranted.

  1. 1. I believe the Abstract needs to be rewritten. It is overly complex and burdensome for the reader. It should be simple and understandable, conveying the importance, purpose, findings, and contributions of the study clearly.
  2. 2."As" should be removed in line 34. Grammar check is needed.
  3. 3.The study feels more like a book chapter than a research article. There is too much basic information in the introduction section. I assume the authors wanted to elaborate on concepts and frameworks in detail. Even the subdivision of the introduction into subsections 1.2, 1.3, and 1.4 might be a consequence of the length of the introduction.
  4. 4.Could the Methodology section be a main heading like 2. instead of a subheading like 1.4? This is something the authors can decide.
  5. 5.The purpose of the study is given in 1.3, but I still feel there are gaps. It's understood that the impact of policy and legal frameworks on ISESs will be examined, but it needs further grounding. I believe there's a similar issue in the Abstract. Questions like gaps in the literature and the study's contribution to the literature remain unanswered.
  6. 6.Have studies or reports on the topic been thoroughly reviewed? The study feels weak in terms of literature. Are there studies examining the social, political, environmental, and economic origins of ISES and ITFS using case studies or other methodologies?
  7. 7. Are there studies empirically examining the mechanisms depicted in Figure 2 and Figure 3 and proving the connections?
Comments on the Quality of English Language

Moderate editing of English language required.

Round 2

Reviewer 1 Report

Comments and Suggestions for Authors

I thank the authors for their thoughtful and detailed responses to my initial review comments, both where they act on those comments and in the cases where they justify not acting on the comments. The revised paper now makes an important contribution to the literature. Well done!

Reviewer 2 Report

Comments and Suggestions for Authors

I can clearly say that the authors have made an effort. Many of my concerns have been addressed. The quality of the work is considered better than the first draft. However, I still have concerns about it being better and as it should be. This work could have been better structured nonetheless. Despite all my reservations, I am making the decision to accept it for publication.

Comments on the Quality of English Language

Moderate editing of English language required.